# Effects of a Short-Term Soccer Training Intervention on Skill Course Performance in Youth Players: A Randomized Study

**DOI:** 10.3390/sports12120345

**Published:** 2024-12-13

**Authors:** Arne Sørensen, Terje Dalen, Pål Lagestad

**Affiliations:** The Faculty of Teacher Education and Arts, Nord University, Høyskolevegen 27, 7600 Levanger, Norway; terje.dalen@nord.no (T.D.); pal.a.lagestad@nord.no (P.L.)

**Keywords:** football, team sport, football skills, children, measurement of skills

## Abstract

The aim of this study was to evaluate the effect of 11 additional soccer training sessions among youth soccer players according to their performance in a skill course. A total of 90 participants, aged 9 to 12, were randomly assigned to either an intervention group (IG) (n = 54) or a control group (CG) (n = 36) and have validated data. The trainings focused upon enhancing ball mastery and decision-making and included a combination of one vs. one situations and small-sided games (SSGs). Pre- and post-tests measured passing and dribbling skills through a skill course. The best time with additional time penalties for each dribbling and passing error was used for further analysis. An independent t-test revealed no significant differences in improvement between the two groups. However, paired t-tests revealed significant improvements for both the IG and the CG from pre- to post-test (7.9 and 3.9 s, respectively). Furthermore, no significant differences in the development of track time, cone touches, or passing errors between the groups were detected. These findings suggest that soccer players aged 9 to 12 improve their performance in a skill course through increased familiarity with the course and natural development of technical skills based on participation in soccer training and recreational soccer play. We argue that the lack of significant differences between the groups’ performances can be attributed to the short duration and few sessions of the intervention, and a somehow low similarity between the skill course and the activities in the sessions.

## 1. Introduction

For players to reach the highest level in soccer, it is crucial to possess exceptional technical and tactical skills. This necessity has heightened the focus on designing the most effective soccer training programs for youth players [1,2]. Therefore, soccer coaches face a challenging task in selecting the optimal training model for their players. According to theories on motor learning, efficient exercises should be specifically related to the training goals and incorporate some variations (contextual interference). The development of technical skills depends on relevant instruction, augmented feedback, and a high number of repetitions [3]. This understanding has led to a popular training method, in which players engage in various closed-loop exercises aimed at developing their isolated technical skills [2]. The belief underlying this method is that as the technical level increases, players will integrate these skills into play activities [2].

Other scholars, however, have been critical of this type of training, arguing that it does not provide players with opportunities to practice decision-making or develop tactical skills [4,5]. Small-sided games (SSGs) or conditioned games have been a popular exercise method in ball games. These games constitute a smaller version of the actual match-situation, with relatively fewer players and smaller areas [6,7]. The advantage of organizing soccer training with SSGs is that it provides players with a multi-faceted approach, enhancing both technical/tactical skills and physical conditioning. Studies have reported moderate effects in interventions evaluating the development of technical skills by using SSGs, even though the differences between the intervention groups and the control groups were small [8,9,10]. In the planning of soccer coaches for the use of small-sided games (SSGs) in training, several factors must be considered, including the number of players, the size of the area, and the playing rules. It is advisable to take into account the aim of the practice and the skill level of the players [10].

Engaging in a significant amount of soccer training and playing between the ages of 5 and 10 can increase the likelihood of becoming a professional soccer player [11,12]. In fact, youth soccer players in some of the largest soccer nations have, by the age of 16, accumulated between 4000 and 5000 total hours of soccer match play and practice [13]. This suggests that youth players must demonstrate a high level of commitment to sport and derive enjoyment from the activities in order to develop their skills to a high level [14]. It is recommended that players with lower skill levels engage in extensive play and practice during their leisure time to enhance their performance [15]. An alternative approach to traditional soccer training, which emphasizes instruction and augmented feedback, has recently garnered significant attention [5,16,17]. This approach, grounded in Nonlinear Pedagogy (NLP), affords young players greater control over their practice, minimizes interruptions from adult “experts”, and could potentially increase their intrinsic motivation for soccer [18]. It has been asserted that a significant portion of soccer training in academies is overly regulated by coaches, which does not support players’ development in a meaningful way to adequately facilitate creativity and player autonomy [16]. Research demonstrated that practicing according to NLP with game-based activities resulted in better development of technical skills than technical drills in a retention test of passing for 10-year-old players, as well as improved decision-making [2,5]. According to NLP, the recommendation for soccer coaches is to organize exercises with the goal of ‘training as you play.’ This means that exercises with low similarity to actual soccer matches should be minimized, and as much time as possible should be dedicated to match-like activities [2].

The development of soccer skills has been demonstrated to constitute a gradual process, characterized by a nonlinear trajectory and significant variability among players [15]. Studies that examined youth players in elite academies indicate that technical skills in dribbling and passing improve with age, exhibiting a gradual increase in skill level until approximately 14–15 years old, after which the progression plateaus [19,20,21,22]. It has also been posited that extensive and relevant practice is a prerequisite for skill development [23]. The specific amount of weekly soccer training of the youth academy players in [19]) and [21] was not described. However, it is reasonable to assume that these players undergo a substantial amount of training. For instance, U14 players in an elite club academy reportedly practiced four times per week on the soccer field, focusing on the development of technical and tactical skills, supplemented by two strength training sessions and one match per week, totaling approximately 10 h of training weekly [24]. The duration of time allocated to playing soccer during leisure periods remains undetermined. This activity, however, serves as a crucial avenue for the improvement of technical skills among young players [13].

In a longitudinal study spanning one year, no significant development was reported in technical skills in dribbling or passing among players aged 12 to 14, despite engaging in 8 to 10 h of training and matches weekly [25]. The authors suggested that the players’ skill levels were relatively high at the study’s onset, thereby limiting the potential for significant improvement [26]. It has been asserted that the duration of interventions is pivotal for performance enhancement. A review examining the impact of small-sided games (SSGs) on the development of technical skills among youth players indicates that interventions with the highest number of sessions (36 and 24, respectively) demonstrated the most significant differences between IGs and CGs in the development of technical skills [6].

It is documented that there is an increase in soccer-specific skills and physical capacities, such as sprinting, endurance, and strength, among soccer players aged 11 to 17. However, there are significant differences among the different age groups due to variability in maturity status [20]. Previous research on skill development has shown that various methods and interventions can lead to positive skill enhancement. The aim of this study was to evaluate an 11-week training program conducted during a soccer school. This evaluation will enhance our understanding of the processes influencing skill acquisition. To elucidate how interventions can affect the improvement of technical skills in soccer, it is crucial to analyze various interventions. This approach allows science to offer research-based guidance on the methods that coaches should choose, ensuring that soccer training is more knowledge-based rather than tradition-based. Previous intervention studies have examined technical skills among adolescents. However, there is a lack of research related to players as young as 9–12 years of age. Therefore, the purpose of this study was to investigate the development of technical skills in a soccer skill course among youth players during an 11-week soccer training program, with one session per week. Moreover, we wanted to assess whether this training program would impact the players’ technical level in dribbling and passing. Our hypothesis is that a short-term soccer training intervention will improve youth soccer players’ performance in a skill course. 

## 2. Materials and Methods

### 2.1. Participants

The inclusion criteria were an application to join a free soccer school and being between the ages of 9 and 12 in the actual community. A total of 175 children, aged 9–12, applied to join the soccer training program during a free soccer school. Of the 175 participants, 100 children were randomly chosen and included in the intervention group (IG), whereas the other 75 were the control group (CG). This took place by placing all 175 names in a bowl and drawing 100 randomly. With such a strategy, the randomization process does not stratify by age, skill level, or other variables to ensure balanced groups. Of the included participants, 90 (54 from the IG (37 boys and 17 girls) and 36 from the CG (23 boys and 13 girls)) were part of the final analysis because they met the criteria for having valid results from both the pre-test and post-test (Table 1). This yielded a response rate of 51.4%. The average participation rate for both groups was also recorded and analyzed. Using power calculations [27] with the numbers from an earlier intervention study with expected differences between groups (1 = 0.39, α = 0.05, β = 0.8) and standard deviation (SD = 0.23), we had to have 53 participants to fulfill the criteria for observed power. This study was approved by the Norwegian Agency for Shared Services in Education and Research (SIKT) on October 24 (2023), with reference number 449551. This investigation was conducted in strict accordance with local legislation and institutional requirements to ensure ethical conduct of research. Prior to commencement of the study, parents provided written informed consent, permitting both themselves and their children to participate in this study. This consent process ensured that all participants were fully aware of the study’s purpose, procedures, potential risks, and benefits.

### 2.2. Training Procedures

The intervention consisted of 11 soccer training sessions of 75 min in the period of November 2023 to February 2024. The selection of 11 sessions was predicated on the organizational structure of the soccer school to which the players belonged. Each supplementary training session was meticulously planned to last for 75 min, with a clear progression mapped out from the first to the eleventh session. Each session was designed with a specific technical or tactical goal in mind, primarily focusing on enhancing ball mastery and decision-making skills. The initial segment of each session predominantly involved 1 vs. 1 situations, with the emphasis placed on the goal of the session. The coaches adopted an instruction-based approach, initially explaining the task to the players, then demonstrating it, and finally allowing the players to apply it in practical soccer situations with minimal defensive pressure. In the second segment, the defensive pressure gradually increased to simulate real match conditions and challenge the players’ skills under pressure. The third and final segment of the training was organized as small-sided games (SSGs), involving activities ranging from 4 vs. 4 to 7 vs. 7. These SSGs were designed to replicate the dynamic and unpredictable nature of an actual soccer match, thereby providing the players with an opportunity to apply their skills in a realistic context. This structured and progressive approach to training was intended to foster skill development and tactical understanding in a supportive and challenging environment. Previous research has demonstrated that both isolated drills and SSG can facilitate skill acquisition among youth players [6]. While a period of practicing SSG has been shown to enhance technical skills, the assessment of tactical skills (decision-making) remains unaddressed due to methodological challenges.

## 3. Testing Procedures and Data Collection

The testing was performed before and after the 11-week training program for both groups. All participants completed a skills course (see Figure 1) twice each at both the pre-test in November 2023 and the post-test in February 2024. Total time in performing the skill course was measured with a stopwatch. Touching the cone with the ball during the dribbling parts of the course (hereafter, cone touch of the yellow or red cones) led to a time penalty of two seconds, and the same time penalty was applied if one missed a pass (hereafter, pass misses). At the start of the skill course (see Figure 1), the participants dribbled through six cones that were placed diagonally two meters apart from each other. Afterwards, the ball had to be stopped within the square at the end of the track (a quadrilateral of 2 m × 2 m), before they had to run, without the ball, between the blue cones. After this, the participants had to shoot passes towards the goal—three passes with the left foot and three passes with the right foot. Once they had shot all the balls, they would move on to the next dribbling part. The ball that was to be used further was placed at the start of the next dribbling part, where the path was set up in a straight line, with two-meter intervals. Finally, the ball needed to be placed in the square and remain still. When the ball was at rest inside of the square, time stopped. All testing was carried out in an indoor soccer facility.

Each player completed the course twice, with a minimum rest period of 10 min between each trial. The test leader closely monitored the players throughout the course, recording each instance when a player touched the cones with the ball, the number of successful passes, and the time taken to complete the course. Participants were instructed to complete the course as quickly as possible. All the children who were going through the skills course were placed in a queue at the starting line. The participants thus had a long break between the two trials. The ready signal to the participants was given with the words “ready, set go”. The selection of the test was influenced in part by the necessity to assess all participants on the same day. The skill course was designed to evaluate critical abilities relevant to soccer matches, specifically dribbling and passing techniques, as well as rapid movement capabilities. However, this test did not assess the players’ decision-making skills.

## 
4. Data Analysis


The best result (track time, cone touches, passing error, and total fails) achieved by each player on the pre-test and post-test was used for further analysis. A variable of penalty time was included in the total track time, with a penalty of two seconds for each cone touched during dribbling and for each unsuccessful pass. This method of measuring passing skills is similar to that used by [28], which could create a more valid test, instead of using the time for fulfilling the course [28]. This is because a soccer player should manage to both dribble with accuracy and speed and make accurate passes. The best time of the two trials (after cone touches and passing errors had been added) was plotted into SPSS version 29 (SPSS, Inc., Chicago, IL, USA) together with the best track times at the pre- and post-tests, number of cone touches, number of passing errors, and total fails. All variables were proven to satisfy the requirements for parametric tests (normal distribution (Kolmogorov–Smirnov > 0.05), meter level, equal variance), and an independent t-test was used to highlight differences between the change in the IG and the CG and during the intervention period. Paired t-tests were employed to examine changes between the pre-test and post-test in both groups according to the best track time, total time, number of cone touches, number of passing errors, and total fails. Independent t-tests were also used to clarify whether there was a difference between the IG and the CG on the pre-test and post-test according to time taken, cone touches, and passing errors. The reported number of training sessions a week between the IG and CG was examined by a Mann–Whitney U test (the variable was not normally distributed (Kolmogorov–Smirnov < 0.001)), and was not significantly different (Z = −0.4, *p* = 0.968). All results were illustrated with descriptive data (mean and standard deviation). The significance value was set at *p* < 0.05. Effect size was evaluated with Cohens d, where 0.2 constitutes a small effect, 0.5 a medium effect, and 0.8 a large effect [27]. 

## 
5. Results


Independent t-tests showed that there were no significant differences between the IG and the CG on the pre-test and post-test according to track time, cone touches, passing errors, and total fails (*p* > 0.05). However, paired t-tests identified a significant decrease between the pre-test and post-test in the IG according to track time (t_52_ = 4, *p* < 0.001, d = 0.549, 95% CI [2.98, 8.98]), total time (t_52_ = 5.8, *p* < 0.001, d = 0.783, 95% CI [5.13, 10.12]), and total fails (t_52_ = 2.4, *p* = 0.021, d = 0.328, 95% CI [0.090, 1.041]). Paired sample t-tests also revealed a significant decrease between the pre-test and post-test in the CG according to track time (t_35_ = 2.5, *p* = 0.017, d = 0.328, 95% CI [0.090, 1.041]), total time (t_35_ = 2.9, *p* = 0.006, d = 0.490, 95% CI [1.18, 6.48]), and passing errors (t_35_ = 3.3, *p* = 0.002, d = 0.551, 95% CI [0.267, 1.121]). 

Track time data, including the penalties for cone touches and unsuccessful passes, are presented in Table 2 [28]. Independent t-tests, however, revealed that there were no significant differences in track time on the pre-test or post-test (*p* > 0.05) between the intervention group (IG) and the control group (CG). The IG had a reduction in total time on the skills course of 6.7 s, while the CG had a reduction in total time of 3.6 s. The reduction in time was thus somewhat greater for the IG, but the independent t-test revealed that there was no significant difference in performance from the pre-test to post-test between the IG and the CG (t = 1.69, *p* = 0.095, d = 0.363, 95% CI [−0.520, 6.335]). 

## 6. Discussion

The primary finding of this study was that both groups increased their performance from the pre- to post-test. Although the IG exhibited a greater reduction in total time on the skills course, this difference was not statistically different from that of the CG. The observed improvement in performance from the pre-test to the post-test for both groups was expected, given the increased age of the participants during the period, which can be explained by natural physical maturing and development of skills during the months from the pre-test to the post-test [19,20]. Some of the improvements in technical performance in both groups could also potentially be attributed to better habituation to the test conditions during the post-test compared to the pre-test, leading to enhanced performance [29]. In addition, this improvement could be due to a better cognitive understanding of how to perform the test, as well as an enhancement in their technical skills. Although youth soccer players develop their technical skills in a nonlinear manner and exhibit individual differences [15], it has been reported that there is an increase in skill level with age until the players reach a plateau at approximately 14–15 years old [19,20,25]. Therefore, the players participating in this study are likely to continue improving their technical skills for several more years. However, this improvement will not occur automatically, but rather will depend on the quality and quantity of their soccer practice [3,23]. 

The finding of no significant difference in the development of technical skills between the IG and the CG aligns with the results reported by [25] in their longitudinal study of youth soccer players aged 12–14. They observed that it takes a considerable amount of practice before any noticeable development can be detected. They also argued that once players have retained a relatively high skill level, further improvement requires extensive training. However, the finding of no significant difference between the groups was somewhat unanticipated, given that the participants were aged 9–12, i.e., an age range typically associated with greater potential for improvement compared to older and more experienced players [19,20]. This principle is described as the Power Law of Practice (PLP) in motoric theory, which posits that the improvement of technical skills is more straightforward at lower levels of proficiency and occurs more gradually as skill levels increase [30]. This finding suggests that the 11 extra soccer training exercises in the IG did not yield an imminent effect, and that a retention test might have demonstrated a significant change [31]. No difference between the groups can also be ascribed to the fact that both groups improved their performance from pre-test to post-test. This increases the performance improvement requirements for the IG compared to the CG for the difference to become significant. 

The intention of the additional training sessions in the intervention was to improve the players’ technical and tactical skills, such as ball mastery in a one vs. one situation, decision-making in position/movement, and passing. The test course employed in this study, however, assessed technical skills in passing and dribbling without evaluating decision-making or tactical knowledge. Furthermore, it has been reported that there is a low correlation between youth soccer players’ technical and tactical skills [32]. Consequently, the IG may have enhanced their tactical skills during the intervention; however, the test utilized in this study did not examine tactical skills. It is plausible that if the 11 additional exercises had included more isolated technical drills and fewer small-sided games (SSGs), the chance for a significant difference between the groups might have been higher. This argument is based on the Identical Elements Theory, which posits that to achieve a positive effect from practicing one skill and expect improvement in another, the elements of these skills must be similar to each other [3]. During the additional exercises, the IG allocated approximately one-third of the time to isolated technical skills. The number of repetitions for each player may have been insufficient for significant technical skill development. Conversely, SSGs have the potential to enhance players’ technical skill development, but this is contingent on the duration of the training period, and it is reported that there are small effects on the development of technical skills during interventions using small-sided games (SSGs) [6]. We argue that 11 extra training sessions for the IG was likely too short a period to expect any difference in technical skills, even for players aged 9 to 12 years. This argument is supported by a longitudinal study reporting no significant development in technical skills in dribbling or passing among players aged 12 to 14, despite engaging in 8 to 10 h of training and matches weekly during a period of 1 year [25]. However, the authors contended that the players’ skill levels were relatively high at the study’s onset, thereby limiting the potential for significant improvement. Moreover, Tillaar et al. [33] found that an intervention with one extra training over a period of 1 year did not exhibit significant importance. It has been posited that the duration of interventions is pivotal for performance enhancement. A review examining the impact of small-sided games (SSGs) on the development of technical skills among youth players indicates that interventions with the highest number of sessions (24 and 36, respectively) demonstrated the most significant differences between intervention groups and control groups in the development of technical skills [6]. The limited duration of the current intervention, comprising only 11 sessions, may account for the absence of significant differences observed in the developmental outcomes between the CG and EG.

It is reasonable to assume that players in both groups participated in soccer training with their clubs and engaged in soccer play during the 12-week intervention. Therefore, the 11 additional sessions constituted only a small portion of their total soccer training during the period. The 11 additional sessions conducted by the intervention group (IG) were guided by an instruction-based approach. Interventions assessing the efficacy of Nonlinear Pedagogy compared to instruction-based methods have demonstrated superior learning outcomes in technical skills with this ‘new’ approach [2,5]. Consequently, if the soccer exercises had implemented Nonlinear Pedagogy, the results could have been significantly different. Furthermore, the sample size could have affected the results, since a larger sample size makes it easier to discern significant differences [33]. Finally, studies have suggested that one extra training session a week is too little to influence performance, even if the intervention occurs over a long period of time [22,25]. 

The results indicated that both groups achieved a significant improvement in their time for completing the skill course, excluding any additional time incurred due to mistakes in passing or dribbling. This improvement is likely attributable to habituation to the test and the natural development of skills within the age group [3,19,21]. Additionally, the results revealed a significant enhancement in passing accuracy for the control group (CG), while the intervention group (IG) showed a marked decrease in their total fails. The improvement in technical skills observed in the IG can be ascribed to the relatively high frequency of the one vs. one situation, including both dribbling and passing situations encountered during the additional training sessions. 

## 7. Strengths and Limitations of the Study

A strength of this study is that it was an experiment with a randomized intervention group (IG) and a control group (CG). To be an intervention study, the study has a large sample size. A weakness of such an intervention with pre- and post-tests could be that changes in performance from the pre- to the post-test can be due to the participants (both IG and CG) having learned and understood the test better, as they have performed the test multiple times by the time they take the post-test. The sensitivity of the chosen skill test could be a limitation. Before testing, the players were instructed on how to perform the test, and the test leader demonstrated its execution. Despite this, some misunderstandings regarding the execution could have occurred. The skill test measured a combination of rapid movement, dribbling, and passing. The results might have differed if an isolated test for dribbling or passing had been used. However, the test–retest reliability of similar dribble tests for children has been reported to be good [34]. The assessment of soccer skills in a manner that ensures both reliability and validity has been a topic of considerable debate within the soccer literature. While our use of isolated tests to measure technical skills can enhance reliability, these methods often suffer from low external validity [29]. Therefore, it has been argued that the measurement of skills should be conducted during actual soccer activities, such as small-sided games (SSGs) [6,22,28]. Future studies examining the efficacy of soccer interventions should use more match-like activities for the testing of skills. 

## 8. Conclusions

The present study evaluated the effect of an 11-week intervention with a weekly session of 75 min training that combined one vs. one situations and small-sided games (SGSs) among participants aged 9 to 12. Our findings showed that both the intervention group (IG) and the control group (CG) showed significant improvements in their performance on the skill course, i.e., 7.9 and 3.9 s improvement, respectively. We argue that this increase is due to increased familiarity with the test course and the development of technical skills based on the players’ participation in normal soccer training and recreational soccer play, since no differences in performance improvement were detected between the groups. Furthermore, no significant differences in the development of track time, cone touches, or passing errors between the groups were detected. There are two plausible reasons for the lack of significant differences in the skill course performance between the two groups: first, 11 sessions may be insufficient to expect substantial improvement; second, the sessions were designed to enhance both technical and tactical skills among the players, whereas the skill course only measured technical skills. 

## Figures and Tables

**Figure 1 sports-12-00345-f001:**
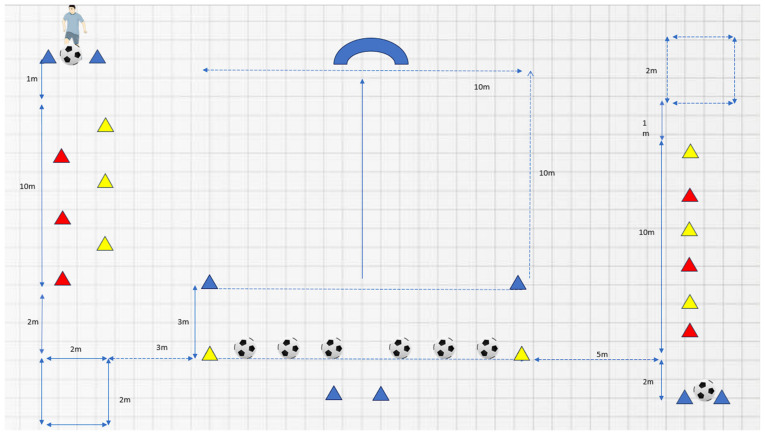
Skill test course, the red, blue and yellow triangles is cones. The small blue squares are the area the players should stop the ball inside.

**Table 1 sports-12-00345-t001:** Number of players in different age groups in the IG and CG.

Age	IG (N)	CG (N)
9 years	25	17
10 years	9	5
11 years	8	3
12 years	12	11

**Table 2 sports-12-00345-t002:** Descriptive data from the pre-test and post-test among the intervention group (IG) and the control group (CG) according to track time, cone touches, passing errors, total fails, and total track time (including penalty seconds, with two seconds for each failure) in the best performance with t values and significance levels from paired t-tests.

Measures	Intervention Group (IG)			Control Group (CG)		
	Pre-Test Mean (SD)	Post-Test Mean (SD)	t Value, *p* Level	Pre-Test Mean (SD)	Post-Test Mean (SD)	t Value, *p* Level
Track time	62.8 (15.6)	56.8 (12.8)	t_52_ = 4, *p* < 0.001	60.8 (12.8)	57.8 (11.9)	t_35_ = 2.5, *p* = 0.017
Cone touch	0.7 (1.0)	0.5 (0.7)	t_52_ = 0.8, *p* = 0.411	0.5 (0.8)	0.9 (1.3)	t_35_ = 3.3, *p* = 0.411
Passing error	3.6 (1.1)	3.2 (1.3)	t_52_ = 1.9, *p* = 0.059	3.5 (1.0)	2.8 (1.0)	t_35_ = 3.3, *p* = 0.002
Total fails	4.3 (1.3)	3.7 (1.5)	t_52_ = 2.4, *p* = 0.021	3.9 (1.4)	3.7 (1.7)	t_35_ = 0.7, *p* = 0.435
Total time	73.4 (15.8)	65.5 (14.3)	t_52_ = 5.8, *p* < 0.001	70.3 (13)	66.4 (12.6)	t_35_ = 2.9, *p* = 0.006

## Data Availability

Data can be obtained by contacting the corresponding author at the following email address: arne.sorensen@nord.no.

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
