# Peer review of "Effects of a Short-Term Soccer Training Intervention on Skill Course Performance in Youth Players: A Randomized Study"

_sports, 2024, doi:10.3390/sports12120345_

Round 1

Reviewer 1 Report

Comments and Suggestions for Authors

The authors need to consider the following comments to improve the manuscript.

The title of the study suggests that it will present evidence of improvements; however, the results show only limited and insignificant changes. Such phrasing may mislead readers, and a more neutral alternative should be considered.

The introduction makes extensive use of existing research, yet it lacks a clear delineation of how this study builds upon or diverges from prior work.

The rationale for selecting an 11-session intervention is unclear and requires further justification. Moreover, I think an 11-week soccer training program comprising a single session per week is inadequate to yield significant findings. It is a very important issue for an intervention study.

It would also be beneficial to ascertain whether external soccer activities (e.g., club training, recreational play) were monitored for both groups, as this could be a significant confounding factor.

The methodology lacks sufficient detail regarding the randomization process for participants, which should be clarified. Was the randomization process stratified by age, skill level, or other variables in order to ensure balanced groups?

It is unclear that comprising a combination of isolated drills and small-sided games was optimized for skill acquisition. Overall it is uncertain whether the test was adequately sensitive to discern alterations in decision-making or ball mastery.

The discussion frequently attributes the absence of notable discrepancies to the brief intervention duration. This is a significant issue for an intervention study.

The conclusion does not adequately address the significance of the study's null results or their implications for future research and practice.

Author Response

Comments from reviewer no. 1                                                       Authors response    

The title of the study suggests that it will present evidence of improvements; however, the results show only limited and insignificant changes. Such phrasing may mislead readers, and a more neutral alternative should be considered.

Thank you for your response. We agree, and have therefore changed the title of the study

The introduction makes extensive use of existing research, yet it lacks a clear delineation of how this study builds upon or diverges from prior work.

Thank you for your response. We have rewritten the article according to the reviewers’ comments

The rationale for selecting an 11-session intervention is unclear and requires further justification. Moreover, I think an 11-week soccer training program comprising a single session per week is inadequate to yield significant findings. It is a very important issue for an intervention study.

Thank you for your response. We have outlined the rationale for the 11-session intervention. While we acknowledge that, ideally, the number of sessions should have been higher, it is also intriguing to assess whether as few as 11 sessions can yield any improvements.

It would also be beneficial to ascertain whether external soccer activities (e.g., club training, recreational play) were monitored for both groups, as this could be a significant confounding factor.

Thank you for your response. We have rewritten the article according to the reviewers’ comments, and training sessions among the two groups have been analyzed and presented in the methods section

The methodology lacks sufficient detail regarding the randomization process for participants, which should be clarified. Was the randomization process stratified by age, skill level, or other variables in order to ensure balanced groups?

Thank you for your response. We have rewritten the article according to the reviewers’ comments, and this is clearer now

It is unclear that comprising a combination of isolated drills and small-sided games was optimized for skill acquisition. Overall it is uncertain whether the test was adequately sensitive to discern alterations in decision-making or ball mastery

Thank you for your response. We have described the assumptions regarding how small-sided games (SSG) can contribute to increased technical skills in the discussion. Additionally, we have clarified that the skill courses were not a measurement of decision-making, but rather a test of ball mastery and rapid movement

The discussion frequently attributes the absence of notable discrepancies to the brief intervention duration. This is a significant issue for an intervention study.

Thank you for your response. We agree and have therefore included in the discussion how the short duration of the intervention affects the likelihood of detecting significant changes between the groups

The conclusion does not adequately address the significance of the study's null results or their implications for future research and practice.

Thank you for your response. We agree and have therefore slightly changed the conclusion based on our own investigation, and added one sentence based on our findings.

Reviewer 2 Report

Comments and Suggestions for Authors

This is an interesting paper about the additional skill training aiming to improve the performance. I will present below my considerations.

The title is inadequate. The title does not reflect the main results of the study.

I suggest further exploring the results in Table 1 in the abstract and the conclusion cannot be supported by the results of the study.

The introduction does not reveal the original aspects of this study. Please be clearer. Please add the hypothesis at the end of the introduction.

The methods section lacks important characteristics of the sample, such as how many were there of each age? Please provide anthropometric characteristics.

The study has a great sample size, however, what were the inclusion and exclusion criteria?

Please add the confidence interval and effect size in all results. The Figure 2 is unnecessary. I suggest remove.

The authors hypothesize that the number of sessions was inadequate to promote improvement in the participants' performance. However, I believe there may be another hypothesis not discussed by the authors. Could it be that playing soccer at this age is not enough to develop skills? I suggest that the authors reflect on this and include it in the discussion.

Author Response

Comments from reviewer no. 2                                                                     Authors response

This is an interesting paper about the additional skill training aiming to improve the performance. I will present below my considerations.

Thank you for your response

The title is inadequate. The title does not reflect the main results of the study.

We agree, and have therefore changed the title of the study

I suggest further exploring the results in Table 1 in the abstract and the conclusion cannot be supported by the results of the study.

Thank you for your response. We agree and have therefore included the results from Table 1 in both the abstract and the conclusion.

The introduction does not reveal the original aspects of this study. Please be clearer. Please add the hypothesis at the end of the introduction.

Thank you for your response. We have incorporated the original aspects of this study into the introduction and have articulated the hypotheses at the end of the introduction.

The methods section lacks important characteristics of the sample, such as how many were there of each age? Please provide anthropometric characteristics.

We have rewritten the article according to the reviewers’ comments, and anthropometric characteristics are presented

The study has a great sample size, however, what were the inclusion and exclusion criteria?

Having valid results from both the pre-test and post-test and a satisfactory participation rate, as highlighted in the methods section.

Please add the confidence interval and effect size in all results. The Figure 2 is unnecessary. I suggest remove.

We have rewritten the article according to the reviewers’ comments, and confidence interval and effect size are presented

The authors hypothesize that the number of sessions was inadequate to promote improvement in the participants' performance. However, I believe there may be another hypothesis not discussed by the authors. Could it be that playing soccer at this age is not enough to develop skills? I suggest that the authors reflect on this and include it in the discussion

We have described in the discussion: “It is plausible that if the 11 additional exercises had included more isolated technical drills and fewer small-sided games (SSGs), the chance for a significant difference between the groups might have been higher.” We have discussed the finding of no significant difference between the group, based on the Identical Elements Theory.  In our opinion, both the type of exercises and the type of test used in this study could have potentially affected the results.

Reviewer 3 Report

Comments and Suggestions for Authors

This study focus on performance improvement and developmental training in youth sports, particularly for soccer players aged 9 to 12. By noting the lack of significant differences between groups due to the intervention’s limited duration and frequency, this study highlights the potential need for longer, more intensive training programs to achieve marked improvements.
This is well written article and the topic is of great significance. The research was conducted according to the principles and rules that apply to longitudinal examination.

The methodology is thorough and facilitates an analysis of the research data.
However, the manuscript has the following deficiencies:

1.In the abstract, Paired t-tests are mentioned first, followed by independent t-tests, while in the rest of the text the order is reversed. The order should be consistent (correct it in the abstract). In the Introduction section, after stating the study objective, the research hypothesis is missing: what did the authors hypothesize?

2.In the Methods and Materials (M&M) section, more details on the sample are needed: did the sample consist of only boys, or were there girls as well, and if so, how many boys and girls were included? What were the inclusion and exclusion criteria? It is stated that of the initial 100 selected participants, 90 met the inclusion criteria. What were these criteria? In the M&M section, explanations are missing in the legend of Figure 1: what do the blue, yellow, and red triangles represent? What do the quadrilaterals (2m x 2m) indicate? What do the solid and dashed lines with arrows signify?

3.In the Results section, the authors refer to Figure 1 but mean Figure 2: "The IG had a reduction in total time on the skills course of 6.7 seconds, while the CG had a reduction in total time of 3.6 seconds (Fig. 1)." Additionally, Figure 2 is entirely uninformative. The significance values mentioned at the end of the following sentence should be indicated and should connect at least the bars in Figure 2: "The reduction in time was thus somewhat greater for the IG, but the independent t-test revealed that there was no significant difference in performance from pre-test to post-test between the IG and the CG (t = 1.69, p = 0.095)";

4.The phrase, "... it has been reported that there is an increase in skill level with age until the players reach a plateau at approximately 14-15 years old..." is repeated several times across different sections. Limit this reference to one instance in the text, along with supporting references 19 and 20. The full terms and abbreviations for the intervention group (IG) and control group (CG) are also repeated several times. This technical issue should be corrected;

5.In the Strengths and Limitations of the Study section, the first sentence clearly states the study’s strengths, but it is unclear what its limitation is. The study limitation should be stated clearly and in simple terms;

6.The same applies to the Conclusion section, which should be concise and clear, directly answering the study’s aim. In contrast, in the last sentence of the Conclusion, the authors reference previous research indicating "that this can be explained by the short duration and too few sessions of the intervention, but also that large standard deviations of improvement from pre-test to post-test affected the possibility of finding a significant difference between the IG and the CG". The conclusion should be based on your own, not on previous research.

Author Response

Comments from reviewer no. 3                                                                   Authors reponse

1.In the abstract, Paired t-tests are mentioned first, followed by independent t-tests, while in the rest of the text the order is reversed. The order should be consistent (correct it in the abstract). In the Introduction section, after stating the study objective, the research hypothesis is missing: what did the authors hypothesize?

According to the consistencey we agree. To ensure consistency, we have made changes in the abstract so that it follows the same order as the rest of the text.

We have also inserted our hypothesis: Our hypothesis is that the intervention group receiving 11 additional training sessions will improve performance more than the control group on our football skills course.

2.In the Methods and Materials (M&M) section, more details on the sample are needed: did the sample consist of only boys, or were there girls as well, and if so, how many boys and girls were included?

What were the inclusion and exclusion criteria? It is stated that of the initial 100 selected participants, 90 met the inclusion criteria. What were these criteria?

In the M&M section, explanations are missing in the legend of Figure 1: what do the blue, yellow, and red triangles represent? What do the quadrilaterals (2m x 2m) indicate? What do the solid and dashed lines with arrows signify?

This information is included in the text now

Being 9.12 years of age, having applicated for the free football school, valid results from both the pre-test and post-test and a satisfactory participation rate, as highlighted in the methods section.

The blue, yellow, and red triangles are cones. The quadrilaterals (2m x 2m) indicate the place where the players stop the ball after finishing one of the dribbling’s. The solid and dashed lines with arrows signify the length of the area. This is clearer in the text now.

3.In the Results section, the authors refer to Figure 1 but mean Figure 2: "The IG had a reduction in total time on the skills course of 6.7 seconds, while the CG had a reduction in total time of 3.6 seconds (Fig. 1)." Additionally, Figure 2 is entirely uninformative. The significance values mentioned at the end of the following sentence should be indicated and should connect at least the bars in Figure 2: "The reduction in time was thus somewhat greater for the IG, but the independent t-test revealed that there was no significant difference in performance from pre-test to post-test between the IG and the CG (t = 1.69, p = 0.095)";

We have rewritten the article according to the reviewers’ comments, and Figure 2 is deleted

4.The phrase, "... it has been reported that there is an increase in skill level with age until the players reach a plateau at approximately 14-15 years old..." is repeated several times across different sections. Limit this reference to one instance in the text, along with supporting references 19 and 20. The full terms and abbreviations for the intervention group (IG) and control group (CG) are also repeated several times. This technical issue should be corrected;

Regarding the first phrase, we have removed one of these sentences so that it is now only referred to once in the introduction and once when we discuss it in the discussion. References 19 and 20 are included both times.

According to the full terms and abbreviations for the intervention group (IG) and control group (CG), we have now reviewed this in the manuscript and changed it so that the abbreviations used are those presented in full the first time in the document.

To ensure that figures and tables are self-explanatory, the entire terminology for the groups has been explained there as well. For intervention groups and control groups generally from other studies, we have used the full term.

5.In the Strengths and Limitations of the Study section, the first sentence clearly states the study’s strengths, but it is unclear what its limitation is. The study limitation should be stated clearly and in simple terms;

In addition to the points we already have mentioned regarding reliability and validity as potential weaknesses, we have also added a point about the athletes could have a better understanding about the test and have learned the different elements of the test better on the post-test.

6.The same applies to the Conclusion section, which should be concise and clear, directly answering the study’s aim. In contrast, in the last sentence of the Conclusion, the authors reference previous research indicating "that this can be explained by the short duration and too few sessions of the intervention, but also that large standard deviations of improvement from pre-test to post-test affected the possibility of finding a significant difference between the IG and the CG". The conclusion should be based on your own, not on previous research.

We agree, and have therefore deleted the last sentence in the conclusion since it is not based on our own investigation, and added one sentence based on our findings.

Reviewer 4 Report

Comments and Suggestions for Authors

I thank the editor for giving me the opportunity to review this manuscript and the authors for the work done. The study aims to investigate the effects of 11 additional soccer training sessions. The trainings focused upon enhancing ball mastery and decision-making and included a combination of 1 vs. 1 situations and small-sided games. The study is intriguing, but I believe there is potential for further improvement. I will outline my suggestions step by step:

The author list is improperly formatted, with missing commas between the authors. Please revise to ensure proper separation of names.

The keywords should be differentiated from the title, this would improve indexing

Abstract 

It is my opinion that the abstract is unclear, the purpose of the study is not clearly stated.

Introduction.

The introduction is well-written; however, enhancing the background and incorporating more literature references could effectively strengthen the manuscript:

Small-sided games (SSGs) or conditioned games have been a popular exercise method in ball games. These games constitute a smaller version of the actual match situation, with relatively fewer players and smaller areas : The benefits of SSGs are mentioned in passing but lack depth

Other studies have shown benefits with similar strategies in the areas of coordination, performance and injury prevention. FIFA 11+ is a warm-up program that is composed of three parts that include 15 exercises, and its application has been primarily focused on preventing and reducing injuries in soccer players. However, by inserting this sequence of exercises into training sessions, some authors have highlighted transversal improvements (Patti, A. et al. Effects of 5-Week of FIFA 11+ Warm-Up Program on Explosive Strength, Speed, and Perception of Physical Exertion in Elite Female Futsal Athletes. Sports 2022, 10, 100. https://doi.org/10.3390/sports10070100)

- Nonlinear Pedagogy (NLP): This concept should be clarified and expanded

- I recommend that the authors review the chronological flow of concepts in the introduction, as some are redundant but not unnecessary. It would be beneficial to present and conclude each concept within a single paragraph before transitioning to the next. At the moment, the central portion of the introduction feels somewhat unclear and disjointed

- I suggest the authors to calculate the sample power with G power or similar.

- Why were the groups not randomized? Please provide justification.

- The authors should include the anthropometric parameters of both groups and assess whether there are significant differences in age, height, and weight.

- Including the analysis of Cohen's d would investigate the effects of the intervention more thoroughly. I would suggest that the authors include this analysis.

-  The authors state that the variables meet the parametric requirements (e.g., normal distribution). However, they do not specify the analyses conducted or the results obtained. This information should be moved to the Results section and presented more clearly.

The discussion is well-written and comprehensive.

Author Response

Reviewer no. 4 has not send any comments

Round 2

Reviewer 1 Report

Comments and Suggestions for Authors

Overall, the manuscript has been improved. For its publication, the authors add some explanations of how playeras as young as 9-12 years of age is different from adolescents in terms of technical skills in the Introduction. Plus, in the table 1, IC need to be changed to IG. 

Author Response

Authors response on Comments from reviewer no. 1, round 2

Overall, the manuscript has been improved. For its publication, the authors add some explanations of how playeras as young as 9-12 years of age is different from adolescents in terms of technical skills in the Introduction

Author’s response:

We have added several sentences at the end of the introduction, regarding the skill levels and physical capacities of young players and described their development with age. Research has shown that there are significant differences within the same age group according to maturity, which affect both soccer skills and physical capacities such as endurance, sprinting, and strength.

Reviewer 3 Report

Comments and Suggestions for Authors

I am satisfied with the corrections made to the paper, but before its publication, I would like to point out the following shortcomings and necessary adjustments:
1. Instead of the cumbersome title "No improvements in skill course performance among youth soccer players – a short intervention study" I suggest the title "Effects of a Short-Term Soccer Training Intervention on Skill Course Performance in Youth Players: A Randomized Study";
2. The authors' hypothesis "Our hypothesis is that the intervention group receiving 11 additional training sessions will improve their performance more than the control group on our football skills course" presented at the end of the Introduction chapter, is not ideally phrased. It is logical that the control group (CG), which does not undergo the intervention, will perform worse than the intervention group (IG). The main hypothesis should focus on the effect of the short-term soccer training intervention on skill course performance in youth players, specifically on the IG participants after the intervention, as the title suggests;
3. In Table 1, the results for the IG should be presented first, followed by the CG, in accordance with the paper’s text, the M&M chapter (subchapter Participants), and Table 2;
4. Add a column to Table 2 showing the significance of the paired sample t-test results (as presented in the text above the table), and accordingly adjust the title of the Table 2.

Author Response

  1. Instead of the cumbersome title "No improvements in skill course performance among youth soccer players – a short intervention study" I suggest the title "Effects of a Short-Term Soccer Training Intervention on Skill Course Performance in Youth Players: A Randomized Study";

Authors response: We have changed the title to the reviewer’s suggestion.

  1. The authors' hypothesis "Our hypothesis is that the intervention group receiving 11 additional training sessions will improve their performance more than the control group on our football skills course" presented at the end of the Introduction chapter, is not ideally phrased. It is logical that the control group (CG), which does not undergo the intervention, will perform worse than the intervention group (IG). The main hypothesis should focus on the effect of the short-term soccer training intervention on skill course performance in youth players, specifically on the IG participants after the intervention, as the title suggests.

Author’s response: We have rewritten the hypothesis according to the reviewer’s suggestion.

  1. In Table 1, the results for the IG should be presented first, followed by the CG, in accordance with the paper’s text, the M&M chapter (subchapter Participants), and Table 2;

Authors response: table 1 is changes according to reviewers’ comments.

  1. Add a column to Table 2 showing the significance of the paired sample t-test results (as presented in the text above the table), and accordingly adjust the title of Table 2.

Authors response: Table 2 changes according to reviewers’ comments.